# Light-Driven H_2_ Production in *Chlamydomonas reinhardtii*: Lessons from Engineering of Photosynthesis

**DOI:** 10.3390/plants13152114

**Published:** 2024-07-30

**Authors:** Michael Hippler, Fatemeh Khosravitabar

**Affiliations:** 1Institute of Plant Biology and Biotechnology, University of Münster, Schlossplatz 8, 48143 Münster, Germany; 2Institute of Plant Science and Resources, Okayama University, Kurashiki 710-0046, Japan; 3Department of Biological and Environmental Sciences, University of Gothenburg, 40530 Gothenburg, Sweden

**Keywords:** H_2_ production, *Chlamydomonas reinhardtii*, electron transfer, CBB cycle

## Abstract

In the green alga *Chlamydomonas reinhardtii*, hydrogen production is catalyzed via the [FeFe]-hydrogenases HydA1 and HydA2. The electrons required for the catalysis are transferred from ferredoxin (FDX) towards the hydrogenases. In the light, ferredoxin receives its electrons from photosystem I (PSI) so that H_2_ production becomes a fully light-driven process. HydA1 and HydA2 are highly O_2_ sensitive; consequently, the formation of H_2_ occurs mainly under anoxic conditions. Yet, photo-H_2_ production is tightly coupled to the efficiency of photosynthetic electron transport and linked to the photosynthetic control via the Cyt *b*_6_*f* complex, the control of electron transfer at the level of photosystem II (PSII) and the structural remodeling of photosystem I (PSI). These processes also determine the efficiency of linear (LEF) and cyclic electron flow (CEF). The latter is competitive with H_2_ photoproduction. Additionally, the CBB cycle competes with H_2_ photoproduction. Consequently, an in-depth understanding of light-driven H_2_ production via photosynthetic electron transfer and its competition with CO_2_ fixation is essential for improving photo-H_2_ production. At the same time, the smart design of photo-H_2_ production schemes and photo-H_2_ bioreactors are challenges for efficient up-scaling of light-driven photo-H_2_ production.

## 1. Introduction

Hydrogen (H₂) production is a promising area of research due to its potential as a clean and sustainable energy source. Among the various biological systems studied for H₂ production, the microalga *Chlamydomonas reinhardtii* has garnered significant interest. *C. reinhardtii* presents several advantages as a model organism for light-driven H₂ production. As a well-established genetic model organism with a fully sequenced genome, it facilitates genetic manipulations and functional studies. Its versatility allows it to grow under various environmental conditions, making it adaptable to different cultivation setups. The alga possesses a highly efficient photosynthetic system capable of harnessing solar energy to drive H₂ production. Notably, it harbors [FeFe]-hydrogenases HydA1 and HydA2, recognized as the most efficient enzymes for H₂ production under anaerobic conditions in nature. However, the primary limitation lies in the extreme sensitivity of HydA1/HydA2 to oxygen (O_2_), which surpasses that of [NiFe]-hydrogenases, hindering the widespread application of *C. reinhardtii* in photo-H₂ production [1].

Green microalgae possess several mechanisms for H_2_ production, leveraging their photosynthetic and metabolic pathways. The primary mechanism is the production of photo-H_2_ through oxygenic photosynthetic electron transfer. In this process, photo-reduced ferredoxin (FDX) could be utilized by HydA1/HydA2 to generate H_2_ [2,3]. The light-driven reduction in FDX is catalyzed by photosystem I (PSI), which is coupled to oxidation of water and production of oxygen at photosystem II. Thus, the generation of photo-H_2_ is always linked to the production of O_2_ and thereby self-limiting (see also below).

Another significant mechanism is photofermentation, where H_2_ is produced under anaerobic conditions during the metabolic fermentation of organic compounds such as sugars or glycerol. This process typically occurs in the light, where microalgae use their enzymatic machinery to ferment organic substrates and simultaneously produce H_2_ as a byproduct [4]. Conversely, dark fermentation involves H_2_ production under anaerobic conditions without light, utilizing stored energy from organic compounds accumulated during light periods. This process relies on the activity of hydrogenases to generate H_2_ from metabolic pathways, offering a versatile means of H_2_ production in the absence of light but requiring prior light exposure for substrate accumulation [4].

Photo-H_2_ production via biophotolysis offers distinct advantages over other mechanisms. It efficiently utilizes light energy to directly split water into hydrogen and oxygen, achieving high energy conversion efficiency. This process is sustainable, relying on renewable resources like sunlight and CO_2_, and enables simultaneous hydrogen production and biomass accumulation, making it a promising avenue for renewable energy generation with low environmental impact.

As photo-H_2_ production is coupled to photosynthetic electron transfer, O_2_ is evolved by PSII, which in turn inactivates the [FeFe]-hydrogenases [5]. Thus, in a natural system, e.g., when Chlamydomonas cells become anoxic due to strong respiring bacterial mats, HydA1 and HydA2 operate as short-lived valves for coping with an excess of energy during abrupt light exposure [6,7]. The finding that Chlamydomonas is producing photo-H_2_ under sulfur deficiency boosted the H_2_ research in the field, followed by advancements in photosynthesis engineering, leading to significant progress in the field in recent years (Table 1) [8]. Yet, photohydrogen production is tightly coupled to the efficiency of photosynthetic electron transport and connected to photosynthetic control at the Cyt *b*_6_*f* complex [9,10,11,12], control of electron transfer at the level of PSII [13], remodeling of PSI [14] and efficiency of linear (LEF) versus cyclic electron flow (CEF) [15]. Consequently, for improved engineering of photo-H_2_, an in-depth understanding of the light-driven H_2_ production via photosynthetic electron transfer is mandatory. Moreover, competition of photo-H_2_ production with CO_2_ fixation and as well as the design novel photo-H_2_ bioreactors need to be addressed. In the following, these aspects are discussed.

## 2. Engineering of Photosynthesis to Boost Photo-H_2_

### 2.1. The Photosynthetic Electron Transfer

In plant photosynthesis, two separate photosystems (PSI and PSII) and an ATP synthase drive light-dependent water oxidation, NADP reduction and ATP formation [40]. ATP is formed at the expense of the proton motive force generated by the light reactions owing to the function of ATP synthase [41]. The Cyt *b*_6_*f* complex links electron transfer between the two photosystems. It functions as membrane-bound plastoquinone (PQ)/plastocyanin (PC) or cytochrome c_6_ (Cyt c_6_) oxidoreductase and also pumps protons into the thylakoid lumen. PC and Cyt c_6_ are oxidized by photo-oxidized PSI. Ferredoxin (FDXs) are reduced as the terminal electron acceptor of PSI. FDX, in turn, reduces the FDX-NADP-reductase (FNR), leading to the formation of NADPH in a process defined as LEF. ATP and NADPH are then consumed by the Bassham–Benson–Calvin cycle for CO_2_ fixation [42]. Alternatively, reduced FDX could reinject electrons into the photosynthetic electron transfer chain in a process called CEF [43] or transfer electrons towards HydA1 and HydA2 (see below).

### 2.2. PSII Dependent O_2_ Production and Photosynthetic Control Associated with PSII Relevant for Photo-H_2_ Production

PSII is a multiprotein complex that catalyzes light-dependent water oxidation, resulting in the production of oxygen [44]. To enhance the light-harvesting and O_2_ production capacity of PSII, various numbers of light-harvesting proteins (LHCB) bind to dimeric PSII core complexes [45]. A structure frequently found in vascular plants and green algae is the C_2_S_2_ supercomplex, in which two copies of the monomeric Lhcb4 and Lhcb5 and two LHCII trimers (S-trimers) are bound to the dimeric core [46]. In vascular plants, larger but less stable PSII supercomplexes, known as C_2_S_2_M_2_, consist of two additional copies of the monomeric Lhcb6 with two additional LHCII trimers (M-trimers) bound via Lhcb4 and Lhcb5 [45,47]. Even larger complexes contain two additional LHCII trimers (L-trimers) bound via Lhcb6 and are referred to as C_2_S_2_M_2_L_1–2_ [48]. A recent study in *C. reinhardtii* identified PSII-LHCII supercomplexes with three LHCII trimers bound to each side of the core (C_2_S_2_M_2_L_2_) [49]. Interestingly, the down-regulation of PSII trimer forming LHCBM proteins, LHCBM1, 2 and 3 in *C. reinhardtii* reduced the PSII antenna size significantly and increased photo-H_2_ production [50], which was also due to diminished to O_2_ mediated HydA inhibition. As mentioned, photosynthetic O_2_ production via PSII is deleterious for H_2_ production as it inactivates the [FeFe]-hydrogenases [5]. To circumvent HydA inactivation and make it more resistant to inactivation by O_2_, mutations were introduced that prevent the interaction of O_2_ with the active center of hydrogenase [16,17,51]. This led to variants that maintain H_2_ production over a longer period of time. Other attempts have been made to change the mechanism of O_2_ production by modifying/down-regulating the PSII O_2_-producing machinery. As mentioned above, treatment of *C. reinhardtii* by nutrient depletion, such as sulfur deprivation [8,15,19,20], led to PSII downregulation and induction of anoxia, followed by sustained H_2_ production. Another way was to genetically modify PSII to allow H_2_ production [21,22,23,24]. The development of new illumination protocols, such as the emission of a series of short light pulses, also enabled continuous H_2_ production [37]. An alternative way to suppress O_2_ concentration in photosynthetic *C. reinhardtii* is the use of an O_2_ scavenger. In a recent paper, a chemical O_2_ scavenger system was utilized on a small scale (10 mL), which increased photo-H_2_ synthesis by 2–5-fold over cells without sulfur deficiency [25]. Similarly, an improved photo-H_2_ production rate was achieved by using an iron-based O_2_ absorbent in the headspace of the culture [26].

Recently, a new type of photosynthetic control associated with PSII was described under anoxic conditions in *C. reinhardtii* [13], reducing maximum photosynthetic productivity threefold. It is hypothesized that photosynthetic control, which depends on the acidification of the lumen pH, leads to a PSII acceptor limitation, which in turn alters the internal electron flow mechanism of the acceptor side, which is reflected in a notable decline in H_2_ production. This is a new concept, as photosynthetic control is known to be mechanistically linked to the Cyt *b*_6_*f* complex, which slows down PQH_2_ oxidation due to lumen acidification to protect PSI. Supporting this hypothesis, a recent study indicated that partial inhibition of electron transport from Cyt *b*_6_*f* to PSI (using small concentrations of DBMIB inhibitor) resulted in periodic surges of H_2_ production, suggesting evidence for the hypothesized photosynthetic control mechanism [12].

### 2.3. Remodeling of PSI Supercomplexes, a Process Relevant for Photo-H_2_ Production

The PSI of green algae can contain up to ten LHCs: two at the PSAL pole and up to eight arranged in two crescents at the PSAF pole [52,53,54,55,56]. On the contrary, high-resolution structures of PSI from vascular plants revealed that it contains four LHCs [57,58,59]. Remarkably, an additional LHCA1-A4 dimer was recently found to bind to the PSAL site in *Arabidopsis thaliana* [60], suggesting that binding of an LHCA heterodimer to the PSAL site also occurs in vascular plants. In this low-resolution complex, an LHCII trimer was also associated and in contact with the additional LHCA1-A4 dimer on the PSAL site. It was suggested that this may represent another PS-LHCI-LHCII state transition complex. This state transition complex is formed when PSII is preferentially excited, and in turn, phosphorylated LHCII proteins detach from PSII to partly connect to PSI (state II). Under conditions in which PSI excitation predominates, this process is reversed. LHCII proteins are de-phosphorylated and associated with PSII (state I) [61]. The kinase responsible for LHCII phosphorylation is STT7 in *C. reinhardtii* [62] or STN7 in *A. thaliana* [63]. High-resolution structures of PSI-LHCI-LHCII complexes were gathered via cryogenic electron microscopy (cryo-EM) from maize [64] and *C. reinhardtii* [65,66]. Interestingly, a *C. reinhardtii* mutant (Stm6), which is blocked in the state 1 transition, showed increased H_2_ production rates under sulfur deficiency [20]. This strain also possessed larger starch reserves, had a higher respiratory rate leading to a low dissolved O_2_ concentration (40% of the wild type (WT) and had less efficient cyclic electron flow (CEF). This all together resulted in H_2_ production rates of Stm6, which were about 10 times higher than the control WT strain.

In a recent cryo-EM study, a PSI dimer was identified as another PSI supercomplex from *C. reinhardtii* [67], where two copies of LHCA9 tether two monomeric PSI in a head-to-head fashion, forming a large oligomeric protein complex. Such dimeric PSI complexes were identified in membranes in dark- and light-adapted membranes of A. thaliana using atomic force microscopy [68]. Similarly, in another study, closely associated PSI-LHCI complexes were identified in solubilized membranes of *A. thaliana* by negative staining electron microscopy [69]. PSI dimers were also identified by cryo-EM analyses of PSI particles from a temperature-sensitive PSII mutant in Chlamydomonas [70].

A reversible PSI dimer formation may have a physiological role in thylakoid membrane structure maintenance in chloroplasts. Importantly, the formation of PSI-LHCI-LHCII and dimerization of PSI are mutually exclusive, as dimer formation would clash with structural features of the state transition complex [67].

Notably, in the dimeric as well as in the state transition but not in the monomeric PSI structure, two loops in the stromal side of PSAG and LHCA9 could be interpreted, likely due to the stabilization by the adjacent monomer in the PSI dimer [67]. This indicates that both PSAG and LHCA9 undergo a structural rearrangement upon dimer and state transitions complex formation, which is absent in the monomer. Steinbeck et al. [71] provided structural evidence that the Cyt *b*_6_*f* binds to PSI towards the PSAG side when LHCA2 and LHCA9 are absent to form a PSI-Cyt *b*_6_*f* supercomplex, underpinning the importance of LHCA2 and LHCA9 for remodeling of PSI (see also below). A model presented in [14] summarized these findings. Notably, the depletion of LHCA2 alters the regulation of photosynthetic electron transfer and hydrogen production in vivo in *C. reinhardtii* [14] (see also below), indicating physiological consequences of PSI-LHCI structural dynamics. In this scenario, depletion of LHCA2 may favor PSI-dimer formation. PSI-dimer formation, in turn, might be more efficient in HydA1 and HydA2 binding and FDX reduction, thereby promoting electron transfer towards hydrogenase. The significance of PSI remodeling for protein–protein interaction and electron transfer between PSI and FDX and between FDX and hydrogenase is depicted in Figure 1. Notably, the PSI-dimer could generate two molecules of reduced FDX at once, which in turn could be used directly to produce a molecule of H_2_ via HydA. Mechanistically, these changes are also linked to PROTON GRADIENT REGULATION5 (PGR5).

### 2.4. CEF and Photosynthetic Control, Competing Processes for Photo-H_2_ Production

Recent work in *C. reinhardtii* provided evidence that PGR5 is involved in CEF [72]. This is in line with its proposed function to facilitate stromal electron transfer into the Cyt *b*_6_*f* [73]. Its deletion thereby strongly disturbs the Mitchellian Q cycle. Also, the deletion of PROTON GRADIENT REGULATION-LIKE1 (PGRL1) impacts CEF in Chlamydomonas [74,75]. It has been proposed that PGRL1 is involved in plastoquinone reduction during CEF [76], yet PGRL1 appears to be rather important for PGR5 expression control and protein stability; in the absence of PGRL1, PGR5 is strongly reduced, mimicking PGR5-dependent phenotypes [74,77,78]. There is evidence that the association of FNR with the thylakoid membrane and its association with PSI supercomplexes is impaired in the absence of PGR5 and/or PGRL1, implying that both proteins, directly or indirectly, contribute to the recruitment of FNR to the thylakoid membrane [79]. These findings suggest that PGR5/PGRL1 knockout-related phenotypes are potentially interconnected to FNR-mediated regulation of photosynthetic electron transport in *C. reinhardtii*, possibly related to the Mitchellian Q cycle during CEF [73,80].

In addition, it is relevant for photo-H_2_ production. The PGR5 deficient strain in the T222 parental background possesses a higher respiration rate and is an efficient H_2_ producer with an average rate with an average rate of about 5 µmol H_2_ mg Chl^−1^ h^−1^ [15]. This *pgr*5 phenotype is similar to the one of Stm6. The integration of the *lhca*2 mutant into the *pgr*5 mutant leads to a double mutant with an even greater potential as an H_2_ producer. The double mutant has been reported as the most potent H_2_ producer under sulfur deprivation [14]. Under sulfur-replete conditions, when the *pgr*5*/lhca*2 strain is simply shifted from darkness to light, it has an initial rate of H_2_ production of 336 μmol H_2_ mg Chl^−1^ h^−1^, significantly higher than the WT and the *pgr*5 mutant [14] and the highest rate measured so far (Figure 2). This corresponds to a light-to-H_2_ conversion efficiency of 10.9%, which close to the 13.4% maximal theoretical conversion efficiency at 413 μmol mg Chl^−1^ h^−1^ [81]. This underpins the assumption, that PSI-dimer formation due to the absence of LHCA2 in the *pgr*5 mutant might further promote photo-H_2_ significantly (see above). It also indicates a great potential to exploit the structural features of PSI in the absence of LHCA2 to enhance photo-H_2_ production.

The finding that photo-H_2_ production is enhanced in the *pgr*5 mutant is likely related to the fact that CEF competes with electron donation to hydrogenases (see below). In line, as outlined above, detachment of FNR from the thylakoid membrane might also impact CEF and/or electron donation to NADP+, which would provide more electrons for photo-H_2_ production. The putative involvement of FNR in CEF has been noted by Iwai et al. [82]. In this work, the isolation of a CEF protein supercomplex composed of PSI-LHCI, LHCII, the Cyt *b*_6_*f* complex, FNR and PGRL1 from state II conditions is described. Functional spectroscopic analyses indicated that this supercomplex performed electron flow under in vitro conditions in the presence of exogenously added soluble PC and FDX [82]. Notably, Terashima et al. [83] also isolated an in vitro active PGR-PSI-Cyt *b*_6_*f* supercomplex of similar composition, also containing FNR, from anaerobic growth conditions. In another work, a low-resolution structure of a PSI-Cyt *b*_6_*f* supercomplex from *C. reinhardtii* isolated from anaerobic conditions was revealed [71]. It is suggested that these PSI-Cyt *b*_6_*f* complexes could be functionally involved in CEF [71]. In this process, FDX might transfer electrons towards Cyt *b*_6_*f,* stimulating CEF between Cyt *b*_6_*f* and PSI. The pathway of CEF shares at least PQ, Cyt *b*_6_*f*, PC, PSI and FDX with that of LEF. Several routes have been proposed for PQ reduction during CEF in microalgae and vascular plants. Besides the PGR5 dependent pathway, CEF also occurs via an NAD(P)H dehydrogenase (NDH)-dependent pathway [84] or direct reduction in a quinone bound to the Q_i_ site of Cyt *b*_6_*f* by combined electron transfer from its proximal heme c_i_ and an FNR bound to the complex [85]. Indeed, recent work [73] suggested PSI and PGR5-dependent stromal electron transfer into Cyt *b*_6_*f* as a part of the Q cycle in algae, as proposed [86]. This functional link between cyclic electron flow (CEF) and electron transfer into Cyt *b*_6_*f* would alter a canonical Q cycle during linear electron flow (LEF) to an alternative Q cycle during CEF [73]. It is currently mechanistically unclear how this electron transfer occurs and how PGR5 is involved. It is suggested that stromal electrons enter the PQ pool directly at the Q_i_ site via bound FNR [85,87]. Thus, the *b*_6_*f* would act as FDX-PQ-reductase as originally suggested by Mitchell [88], and FNR would tether FDX during this process, which could drive the reduction in heme-*c*_i_. The operation of the Q cycle [86] contributes to both ΔpH and ΔΨ formation. In turn, elevated acidification of the lumen slows down the oxidation of PQH_2_ at the Cyt *b*_6_*f*, leading to a slowdown of overall photosynthetic electron transfer, designated as photosynthetic control [10]. Interestingly, in *C. reinhardtii,* photosynthetic control is particularly established under anoxic CEF-promoting conditions [9]. Thus, under anoxia, electron transfer towards photo-H_2_ is limited by acidification of the lumen driven by the Mitchellian Q cycle, which consumes electrons from FDX, the electron donor for HydA1 and A2, and slows down the rate-limiting step of photosynthetic electron transfer, which is the oxidation of PQH_2_ at the Qo site of the Cyt *b*_6_*f* [89]. Now, the following question arises: why is photo-H2 elevated in the absence of PGR5? Here, two reasons appear to be plausible; one, as already mentioned, is that in the absence of PGR5, fewer electrons are directed toward CEF, and more are available for HydA1 and A2. The other reason could be related to an impact in the Mitchellian Q cycle, which leads to a less acidified lumen, thereby hindering the full onset of photosynthetic control. In this way, and under defined anoxic conditions, more electrons flow to PSI and FDX, which in turn might enhance electron transfer toward photo-H_2_ production.

## 3. Engineering of Photosynthesis in *C. reinhardtii* to Boost Photo-H_2_—Carbon Fixation

Hydrogenase enzymes must compete with different metabolic pathways to acquire photosynthetic electrons from FDX, which is the main electron hub. Carbon fixation by the Calvin Benson Bassham (CBB) cycle, as the strongest sink of electrons, is the most robust competitor [90,91]. Due to the high demand for NADPH by the Calvin cycle and the fact that hydrogenase is a significantly less efficient electron acceptor than FNR, the Calvin cycle outcompetes H_2_ production within two minutes of the transition from dark to light [92,93]. Contrary to the earlier dogma [94,95], now it is established that the inadequate competitiveness of hydrogenase against the Calvin cycle for photosynthetic electrons is the primary cause for the halt in H_2_ production, occurring even before the inactivation of HydA by oxygen resulting from water splitting [7]. Moreover, it is reported that the heightened activity of the CBB cycle likely leads to a shifting of HydA activity towards H_2_ uptake. This is suggested to be a physiological response to the high demand of the CBB cycle for NADPH under microoxic conditions and at light, wherein hydrogenase releases electrons from H_2_ for use in NADP+ reduction [96]. It can serve as a protective mechanism for cell survival; however, it is not ideal from the biotechnological perspective, as this H_2_ uptake hinders the potential of sustainable H_2_ production.

In total, the challenge of insufficient electron allocation to HydA predominantly arises mainly from electron loss to the CBB cycle, and thus impairing CO_2_ fixation can be considered as a promising solution to boost H_2_ production. The established strategies to manipulate the electron partitioning between CO_2_ fixation and H_2_ production can be categorized into two main approaches: (1) synthetic biology-based approaches and (2) designing new protocols. All these strategies aim to redirect photosynthetic electron transfer to improve electron allocation to HydA.

### 3.1. Synthetic Biology Approaches to Enhance Electron Supply to HydA

Genetic engineering of various targets to divert electron flow to HydA has shown varying degrees of improvement in both the duration and rate of H_2_ production. In earlier studies, the ribulose-1,5-bisphosphate carboxylase/oxygenase (Rubisco) enzyme was considered an engineering target for impairing the CBB cycle in *C. reinhardtii*. The introduction of a missense mutation into the large sub-unit of Rubisco resulted in a higher capability of H_2_ production of the resulted mutant compared with the WT parent. But it also enhanced the extent of photoinhibition of PSII [33,34]. Alternatively, the knock-down of Rubisco was examined by point mutagenesis in tyrosine 67 of the Rubisco small subunit. The resulting mutant (Y67A) was able to grow in low light conditions, with significantly improved hydrogen productivity under sulfur deficiency [35]. The light sensitivity of the Rubisco mutants implies that a certain level of CBB cycle activity is crucial for repairing the photodamaged PSII, maintaining the integrity of photosynthetic machinery, and thereby achieving sustainable H_2_ production [34]. Another interesting mutant in this regard is the temperature-sensitive mutant TSP2, which has a high H_2_ production phenotype when grown at 37 °C (12 μmol H_2_ mg Chl^−1^ h^−1^) [97]. It harbors a single amino acid change Pro160Leu in Phosphoribulokinase (PRK) [36]. At 37 °C, the enzyme is inactive, but its quantity does not change, which indicates that the point mutation makes the enzyme temperature-sensitive. As PRK catalyzes the phosphorylation of ribulose 5-phosphate by the use of ATP, the CBB cycle does not work in the absence of active PRK. These results also point to a strong competition between CO_2_ fixation and the CBB cycle.

FDX1, also known as PETF, presents an alternative target for engineering electron transport redirection towards HydA. Six out of twelve ferredoxin isoforms have been characterized in *C. reinhardtii*. And only two out of the six chloroplast-localized ferredoxins, known as FDX1 and FDX2, are functionally linked to the hydrogenases [98]. The capability of FDX2 to provide electrons to the hydrogenase is less than half of that observed for FDX1. Thus, FDX1 is considered the primary electron donor to the hydrogenase [92,99]. However, the affinity of FDX1 for FNR is 4- to 13-fold higher than that for HydA1 (Km (FNR) = 0.8–2.6 mM, Km (HydA1) = 3.4–35 mM) [100]. Moreover, on the protein level, FNR is about 70-fold more abundant than HydA1 in anoxic *C. reinhardtii* cells [101]. Therefore, a strategy for enhancing H_2_ photoproduction by *C. reinhardti*i has been to engineer proteins aimed at increasing the interaction of FDX1 with HydA1 rather than FNR.

Two conserved aspartic acid residues, D19 and D58, within FDX1, were found to play a crucial role in the selective recognition of as an electron transfer binding partner [32], while they do not affect the interaction of FDX1 with HydA1 [98]. Site-directed mutation of these residues led to a decreased affinity of FDX1 for FNR, likely redirecting photosynthetic electrons toward HydA1 and consequently enhancing in vivo hydrogen production fourfold [32].

Another effective strategy for enhancing electron transport to HydA involved fusing the FDX1 protein to HydA. The FDX1-HydA1 fusion protein was shown to redirect over 60% of the photosynthetic electron pool towards H_2_ production in vitro, a significant increase compared to the less than 10% observed for natural HydA1 protein [91]. Notably, in vivo expression of this fusion protein in *C. reinhardtii* resulted in a 4.5-fold improvement in production compared to the wild type [102]. This enhancement was attributed to the tethering of FDX1-HydA1 to PSI, which facilitated the diversion of electron flux towards HydA. Additionally, the fusion protein exhibited improved O_2_ tolerance both in vitro and in vivo, retaining 25% activity after a 10 min exposure to O_2_ (1.7 µmol), compared to only 7.5% activity for wild-type HydA1. Although the molecular mechanism behind this improvement requires further investigation, it is hypothesized that the FDX1 moiety of the fusion protein either reduces O_2_ to O_2_^−^ or partially blocks O_2_ from accessing the active site of HydA1 [102].

Rewiring photosynthesis to deliver electrons from PSI directly to HydA also has shown promise, both in vitro [29,30] as well as in vivo, for photo-H_2_ production by *C. reinhardtii* [100]. The first direct fusions of HydA to PSI were accomplished in vitro by self-assembly of genetically modified and separately purified proteins [29]. In vivo fusion of PSI and HydA was achieved by inserting the HydA sequence into the PsaC subunit of PSI, resulting in a co-assembled active photosystem. Cells expressing only the PSI-hydrogenase chimera indicated high rates of photo-H_2_ production for several days [100]. It was later revealed that HydA activity in this fusion protein could be restored after complete inactivation by O_2_, indicating that the active site of the enzyme could be reinserted into the same PsaC-HydA1 protein by hydrogenase maturases [31]. Surprisingly, this recent study disclosed that the PSI-HydA1 chimera could also reduce ferredoxin in vivo to a degree that drives the CBB cycle, although FDX1 reduction was hampered overall. This led to high O_2_ production rates and eventually inactivating the hydrogenase, making this mutant unsuitable for long-term sustainable H_2_ production [31].

In another study, a fusion protein combining HydA1 and superoxide dismutase (SOD) significantly boosted H_2_ production [18]. Although originally designed to enhance the O_2_ tolerance of HydA1, the fusion protein did not protect HydA1 from O_2_ but sustained continuous H_2_ production for up to 14 days. It achieved the average H_2_ production rates of up to 10–15 µmol H_2_ L^−1^ h^−1^ in *C. reinhardtii* without the need for any nutrient deprivation, using a standard TAP medium. This prolonged and enhanced H_2_ production is attributed to the ability of fusion protein to compete with the CBB cycle for electrons. However, this competition with the CBB cycle had some drawbacks, as *C. reinhardtii* HydA1-SOD mutants exhibited slower growth rates, likely due to impaired photosynthetic activity [18,34]. The molecular mechanism behind this great competitiveness remains unclear, but it is hypothesized that the HydA1-SOD fusion protein binds to (or in close proximity to) PSI, limiting FNR’s access to reduced FDX. In an alternative hypothesis, HydA1-SOD may outcompete soluble FNR for electrons in the stroma [18].

Structural and functional remodeling of PSI, established in *pgr*5/*lhca*2 double mutant, has also been revealed as an interesting and promising strategy for rerouting electrons towards HydA. Although the estimated electron transport rate in the double mutant was similar to that of the wild type, the rate of CO_2_ assimilation was significantly lower, proposing a redirection of electrons towards hydrogenase or possibly towards O_2_ photo-reduction. It is speculated that in this double mutant, PSI-dimerization supports H_2_ production either via recruiting HydA to PSI and/or by displacing FNR, resulting in an exceptional H_2_ production rate [14] (see also above).

### 3.2. Customized Methods of Photo-H_2_ Production to Enhance Electron Supply to HydA

In addition to genetic engineering, promoting electron allocation to HydA can be achieved by developing new methods and protocols for algal photo-H_2_ production. Nagy et al. [26] proposed a method based on fully limiting the substrates for the CBB cycle to block carbon fixation. Green microalgae, such as *C. reinhardtii,* can use both CO_2_ and acetate as carbon sources for the CBB cycle. Acetate assimilation occurs via the tricarboxylic acid cycle and the glyoxylate cycle, which are linked to gluconeogenesis and the oxidative pentose phosphate pathway. The CO_2_ and glycerate 3-phosphate are released and then fed to the CBB cycle [103,104]. By omitting both CO_2_ and acetate and implementing some previously established measures, this method effectively boosted H_2_ production. The protocol involves a short anaerobic dark incubation of the cultures to induce HydA expression, followed by continuous high light intensity incubation (320 µmol photons m^−2^ s^−1^) in an acetate-free medium without CO_2_ supply and daily N_2_-flushing of the culture headspace. The absence of CBB cycle substrates, combined with an iron-based O_2_ absorbent to maintain hypoxia, sustained H_2_ production for several days, yielding higher cumulative H_2_ production compared to the sulfur deprivation protocol [26]. In this system, the *pgr*5 mutants showed excellent performance and proved to be well-suited for both sunlight intensity and varying light conditions [27].

Pulse illumination protocol is another novel method developed by [37] for bypassing the competition of CO_2_ assimilation with H_2_ production. This approach is based on a simple light-paradigm shift from continuous illumination to a series of white light pulses (1 s) interrupted by longer dark phases (9 s). The short illumination period prevents activation of CBB cycle enzymes, while the dark period allows oxygen removal through respiration, activating HydA. This method sustained H_2_ production for three days, achieving a maximum specific rate of 25 µmol H_2_ mg Ch^−1^ h^−1^ [37]. Recently, improvements to this protocol, such as increasing the illumination period to 2 s and using red light (660 nm) instead of white light to boost the electron source, were proposed [38]. They also introduced low concentrations of sodium sulfite, a proven oxygen scavenger [39], into the culture to eliminate evolved O_2_ from the increased light period. Among the wavelengths tested, 660 nm red light was optimal for H_2_ production, probably due to decreased gene expression of Rubisco and FNR [38].

By combining the light/dark cycle protocol (2 min light and 3 min dark) with the omission of CBB cycle substrate (both CO_2_ and acetate), Milrad et al. could achieve an average production rate of 49 µmol H_2_ mg Chl^−1^ h^−1^ under an irradiance of 370 µmol photons m^−2^ s^−1^ in the short term [7].

Alongside these established protocols, meticulous optimization of cell incubation conditions to favor electron partitioning to HydA can significantly boost H_2_ production yield. During the cell growth phase, *C. reinhardtii* cultures are typically incubated at room temperature. However, transitioning the cultures to an increased temperature of 34 °C at the beginning of the H_2_ production phase, as proposed by [105,106], can enhance H_2_ production efficiency when combined with other effective strategies. In this work, it was shown that photo-H_2_ can be produced under mixotrophic conditions and higher temperatures in the *pgr*5 mutant. This revealed a novel protocol where no nutrient deficiency is required for photo-H_2_ production [105,106]. Higher temperatures reduce the solubility of gas molecules, allowing more evolved H_2_ to escape from the liquid phase to the gas phase, thereby bypassing H_2_ uptake and reducing electron loss to the CBB cycle [105,106]. Additionally, CO_2_ is less soluble at higher temperatures, making it less available for the CBB cycle [107]. Moreover, temperatures above 20 °C are likely to induce reversible inactivation of CO_2_ fixation [108]. Therefore, adjusting the incubation temperature could serve as a switch for activating or deactivating CO_2_ fixation, fine-tuning the H_2_ production process [107].

## 4. Engineering of Photo-H_2_ by Tailoring Photobioreactor Design

To develop an economically viable system for green H_2_ production using green algae, it is crucial to engineer photobioreactors (PBRs) specifically tailored for outdoor photo-H_2_ production. The ideal PBR should support high cell density achievement to maximize productivity per surface area, and it should also facilitate optimal sunlight-to-H_2_ conversion efficiency [109,110]. Additionally, the regular collection of produced H_2_ (to avoid H_2_ uptake) is an essential requirement of an effective PBR for photo-H_2_ production [110,111]. According to Burgess et al. (2006), the efficiency of H_2_ collection is influenced by several factors, including the materials used in the reactor (H_2_ permeability coefficient), the geometry of the reactor (such as wall thickness, reactor diameter and length of joints), as well as the velocities and volume ratios of the gas and liquid [112]. Key design considerations for an ideal photo-H_2_ production PBR include the following: (1) the optimal selection of the material and geometry; (2) effective agitation methods; (3) a robust gas discharge and collection system; (4) airtight connections to prevent ambient O_2_ intrusion; (5) compatibility with the requirements of the most promising established protocols for photo-H_2_ production [113,114].

Scaling up of different protocols initially developed in lab-scale setups can encounter various challenges that significantly impact the efficiency of photo-H_2_ production. For instance, an attempt by Scoma et al. [115] to produce H_2_ with sulfur-deprived *C. reinhardtii* in a 50 L outdoor tubular bioreactor failed to replicate laboratory-scale results. The light-to-H_2_ energy conversion efficiency in their tubular pilot-scale PBR was only about 18% of what was achieved in the lab. Several factors contributed to this discrepancy, including differences in PBR geometry and mixing systems, turbulence rates, light intensity and uniformity [115]. Additionally, the conventional method of sulfur deficiency used in this pilot trial is not an efficient strategy for sustainable H_2_ production.

Later studies have confirmed that among various PBR configurations, flat panel formats are the leading choice for photo-H_2_ production, primarily due to their ease of construction and high surface-to-volume ratio, which result in superior light distribution [113,116]. These features make them effective for large-scale cultivation, outperforming tubular PBRs in terms of photosynthetic efficiency [117]. It has been shown that these benefits can be harnessed with a suitable design for producing photo-H_2_ through well-established protocols. The flat plate culture vessel can be positioned horizontally with a rocking motion agitation, which creates a large surface area between the culture and headspace, even with a small headspace volume inside the reactor. This feature provides an advantage for the release and collection of hydrogen gas compared to bulk PBRs [114,118].

Adapting these considerations, Nagy et al. designed a thin cell layer photobioreactors (TCL-PBR), when instead of traditional bulk culture flasks, they utilized a 1 L flat culture flask including a dense thin layer culture, along with a fabricated O_2_-absorbant holder in top [28]. With this PBR, they managed to enhance the H_2_ productivity attainable by substrate-limitation protocol approximately three-fold as compared to that achieved using traditional bulk. Interestingly, this PBR setup enabled continuous H_2_ production at sunlight intensity (1000 µmol photons m^−2^ s^−1^) when cell density was increased from 50 to 150 µg Chl/mL. Among the four strains of *C. reinhardtii* tested, the *pgr*5 mutant showed exceptional performance, maintaining its photosynthetic apparatus and HydA activity for six days, achieving a total of 53 µmol H_2_ mL^−1^ [28]. This impressive H_2_ production rate under sunlight intensity represents a significant step towards outdoor H_2_ production.

However, another important consideration is how this system performs under daily light cycles and variable light intensities. In a recent study by Nagy et al., the TCL-PBR was equipped with an automated system to continuously monitor H_2_ production in thin-layer cultures of green algae under simulated daily light conditions [27] (16 h light/8 h dark and under variable light intensities between 0 and 1000 μmol photons m^−2^ s^−1^). This automated PBR featured regular N_2_ flushing, pressure control and automated gas sampling. The total H_2_ yield obtained by the *pgr*5 mutant under dark–light cycles [27] was lower compared to continuous illumination following a short dark incubation period [105,106]. However, dark–light cycles offered an advantage for the *pgr*5 mutant. In continuous light, daily H_2_ production decreased significantly towards the end of the experiment, while in dark–light cycles, H_2_ production remained more stable. This stability may be due to partial regeneration of HydA during dark periods [27]. Another type of cyclic photosynthetic hydrogen-producing bioreactor [119] has been introduced to take advantage of temperature-sensitive mutants such as the TSP2 mutant [36,97]. Accordingly, in this reactor, the *C. reinhardtii* culture circulates between two illuminated main chambers at different temperatures. At 25 °C, the culture grows in an aerobic bottle; at 37 °C, the culture produces hydrogen in a sealed anaerobic collection chamber.

Furthermore, varying light intensity (0 to 1000 μmol photons m^−2^ s^−1^) positively impacted H_2_ production in both the CC-124 strain and the *pgr*5 mutant. This effect might be due to HydA’s role in relieving pressure on the photosynthetic electron transport chain. Electrons are likely transferred to HydA more efficiently as light intensity increases. Under these simulated outdoor conditions using the automated TCL-PBR, the *pgr*5 mutant was confirmed to be a promising candidate for H_2_ production in bio-industrial settings, even under variable conditions [27].

Recently, Chen [114] proposed a concept design for large-scale photo-H_2_ production by integrating the dark/light cycle protocol with the substrate limitation protocol (Figure 3). This approach appears promising and economically feasible, though it requires proof of concept. The design features a two-layer structure: the top layer photobioreactor is exposed to sunlight for H_2_ production, while the bottom tank is kept in darkness for HydA enzyme recovery. A water pump regulates the flow rate to maintain a cycle of 2 min of light and 3 min of dark. The bottom tank includes a degasser to separate H_2_, and its larger size aids in the release of aqueous H_2_ due to hydraulic pressure changes at the interface between the top tube and the tank. One notable advantage is that many algal species can grow under mixotrophic conditions, allowing growth during the nighttime (with a limited substrate supply) and H_2_ production during the daytime [114].

## 5. Conclusions

This review underscores the remarkable potential of *C. reinhardtii* as a model organism for photo-H_2_ production. Recent advancements in engineering photosynthesis, alongside the development of various protocols and PBRs, have significantly enhanced the achievable H_2_ productivity by this alga. Among the numerous mutants of *C. reinhardtii* engineered to improve H_2_ production efficiency, the pgr5 and pgr5/lhca2 mutants have emerged as leading H_2_ producers.

To fully exploit the capabilities of these mutants for photo-H_2_ production, future research should focus on gaining a deeper understanding of photosynthetic control mechanisms under anaerobic conditions and optimizing these processes to favor H_2_ production. Moreover, the link between CO_2_ fixation efficiency, respiration and photo-H_2_ production needs more attention.

Additionally, integrating photo-H_2_ production with the generation of other valuable algal products represents another promising area for further research. This approach, by enabling the production of multiple products from a single algal biomass, could enhance the economic viability of algal H_2_ production.

## Figures and Tables

**Figure 1 plants-13-02114-f001:**
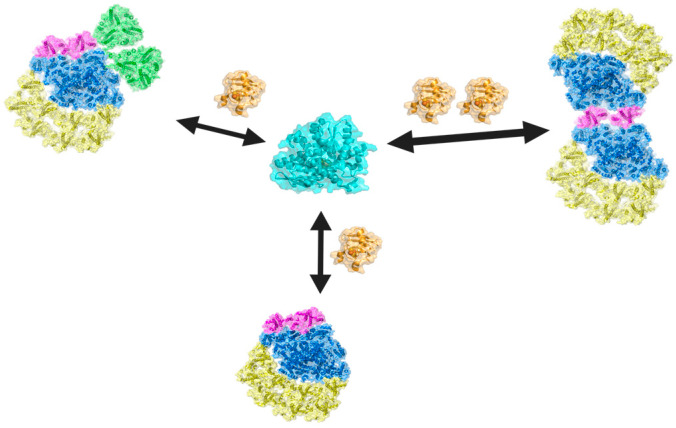
Schematic view on PSI remodeling processes and electron transfer between FDX (brown) and HydA (cyan). The monomeric PSI-LHCI complex (PSI core. Blue, LHCI belt at PSAF side, yellow, LHCA9-LHCA2 hetero-dimer, pink) can be remodeled into the state transition complex, with two LHCII-trimers (green) or the PSI-dimer, tethered head-to-head via LHCA9. The PSI dimer can photo-reduce two molecules of FDX (brown) at once, which in turn could be used directly to produce a molecule of H_2_ via HydA (cyan). HydA could also bind to the various PSI complexes, with binding to the PSI dimer being favored. PDB: 7DZ7: 7ZQC: PSI-monomer, PSI-monomer+ LHC trimers, 7ZQD: PSI-dimer 6KV0: FDX1 (*Chlamydomonas reinhardtii*), 6N59: HydA (Clostridium).

**Figure 2 plants-13-02114-f002:**
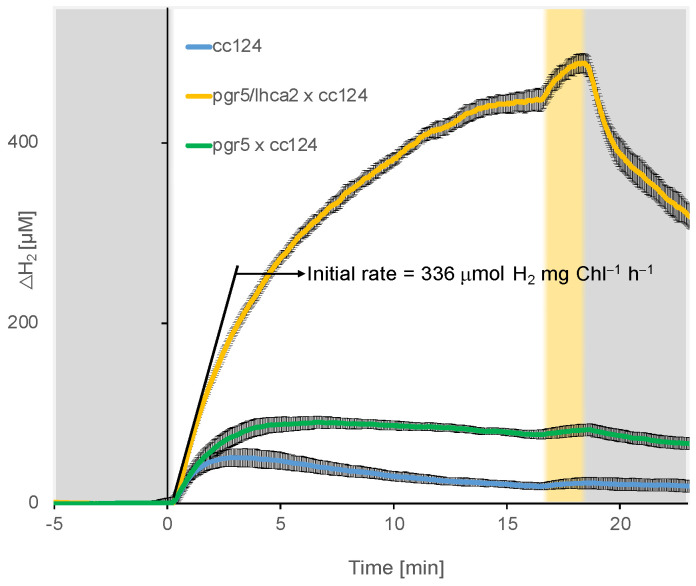
Short-term kinetics of dissolved H_2_ measured by membrane-inlet mass spectrometry (MIMS). cc124, *pgr*5/*lhca*2 and *pgr*5 cells at a concentration of 15 µg Chl mL^−1^ were incubated in the dark for 2 h (in Tris–Acetate–Phosphate (TAP) medium), after which they were exposed to 16 min of illumination (370 µmol photons m^−2^ s^−1^; white background) followed by 2 min of high light (2500 µmol photons m^−2^ s^−1^; yellow background) (modified from [14]).

**Figure 3 plants-13-02114-f003:**
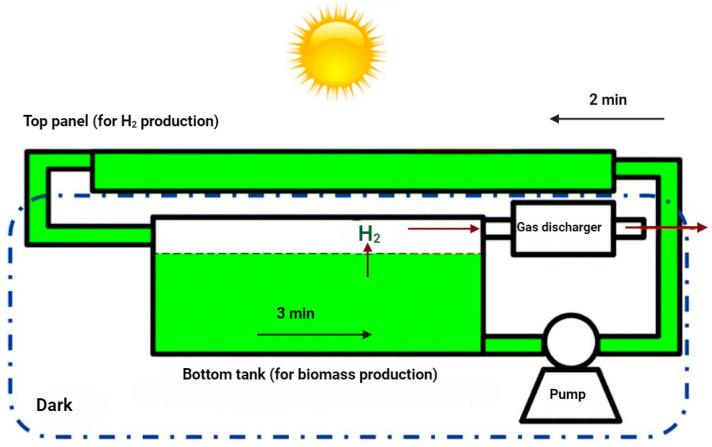
A large-scale design concept integrating the 2 min light/3 min dark cycle method with the substrate limitation approach [114].

**Table 1 plants-13-02114-t001:** Key factors for photo-H_2_ production in *Chlamydomonas reinhardtii*.

Key Proteins	Involving Role	Effective Modifications to Boost H_2_ Production	Achievement	Reference
HydA1/HydA2		Manipulating the active site of the enzyme to decrease interaction with O_2_	Improved O_2_ tolerance	[16,17]
	Catalyze production of H_2_ from electron and proton	PSI-HydA1 fusion	Deliver more electrons to HydA	[18]
PSII		Nutrient deprivation	Gradual inhibition of PSII activity and establishment of hypoxia	[8,15,19,20]
	Generate e- and H+ for HydA	Genetic modification of PSII subunits	Down-regulation of PSII activity	[21,22,23,24]
	O_2_ evolution and HydA activity inhibition	Use of O_2_ absorbents	Establishment of hypoxia while preserving PSII activity	[25,26]
Cyt *b*_6_*f*	Regulate photosynthetic electron transport based on redox state of thylakoid membrane	Down-regulation of electron transport from Cyt *b*_6_*f* to PSI	Regulating H_2_ production by adjusting the redox state of the thylakoid membrane	[12]
PGR5	Mediate CEF Regulating the rate electron transport to HydA Regulating photo-protective mechanisms	CEF-deficient mutants (*pgr*5 and *stm*6)	Higher respiratory rate Higher stability of PSII More electron allocation to HydA	[15,25,27,28]
PSI	Electron transfer to FDX	Putative PSI dimerization (in pgr5/lhca2 mutant)	More efficient electron transport to HydA	[14]
		PSI-HydA1 fusion	Deliver more electrons to HydA	[29,30,31]
FDX1	Final electron donor to HydA	Point mutation of FDX1 to decrease the affinity for FNR	More efficient electron transport to HydA	[32]
CBB cycle enzymes	Competing with HydA for photosynthetic electron	Mutation of Rubisco sub-units	Partial improvement of electron delivery to HydA (but more vulnerable to photoinhibition)	[33,34,35]
		Temperature-sensitive mutant of PRK	Less activity of CBB cycle at 37 ℃	[36]
		CBB Cycle substrate limitation	Limitation of CBB cycle activity due to CO_2_ and acetate starvation	[26,27]
		Pulse illumination	Preventing activation of CBB cycle enzymes due to very short light periods	[37,38,39]

## Data Availability

Not applicable.

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
