# Peer review of "Light-Driven H2 Production in Chlamydomonas reinhardtii: Lessons from Engineering of Photosynthesis"

_plants, 2024, doi:10.3390/plants13152114_

Round 1

Reviewer 1 Report

Comments and Suggestions for Authors

In this manuscript (plants-3120973) entitled "Light-driven H2 production in Chlamydomonas reinhardtii – lessons from engineering of photosynthesis" submitted to Plants, Michael Hippler and Fatemeh Khosravitabar have discussed the light-driven H2 production in Chlamydomonas reinhardtii and its application in photo-H2 production. This review is interesting and timely, but this present manuscript needs revisions before publication.

Major points:

1. The introduction section is too short. Author should consider to expand this section in the revision.

2. A table summarizing key proteins affecting light-driven H2 production in Chlamydomonas reinhardtii should be included in the revision.

3. A section of conclusion should be included in the revised manuscript.

4. Figures presented in this manuscript were modified from published work. Authors should consider to show more original figures in the revision.

Minor points:

1. Authors need to standardize references according to the Plants template. For instance, abbreviation instead of full name of Frontiers in Plant Science should be presented (Reference 12).

Author Response

We would like to thank the reviewer for the positive and critical feedback on our manuscript.

Reviewer 1.

  1. The introduction section is too short. Author should consider to expand this section in the revision.

We expanded the introduction.

  1. A table summarizing key proteins affecting light-driven H2productionin Chlamydomonas reinhardtii should be included in the revision. 

A table, summarizing key proteins affecting light-driven H2 production in Chlamydomonas reinhardtii was added.

  1. A section of conclusion should be included in the revised manuscript.

A conclusion section was added.

  1. Figures presented in this manuscript were modified from published work. Authors should consider to show more original figures in the revision.

Figure 1 was revised and newly done. Figure 2 shall be kept, as it displays the rate of H2 production in pgr5/lhca2 that was not calculated in Ho et al. (2022). Figure 3 was newly made using the old one as template.

Minor points:1.     Authors need to standardize references according to the Plants template. For instance, abbreviation instead of full name of Frontiers in Plant Science should be presented (Reference 12).

Done

Reviewer 2 Report

Comments and Suggestions for Authors

Dear Authors,

In this article, the authors review the latest advances in light-dependent H2 production in Chlamydomonas. I believe the review is very well-written; the authors are of recognized prestige, which is evident in how well they discuss and handle the subject matter. However, I miss some ideas that I think should at least be reflected in the review. I also suggest including a table, which would improve the impact of the review. There is no conclusion section; it would be very beneficial to include one that addresses the main biotechnological barriers faced by microalgae H2 production.

Majors:

-The introduction seems short to me; at the very least, it should mention that Chlamydomonas has other mechanisms of H2 production that are independent of light  that are independent of PSII and can occur in the dark.

- The introduction should be improved, particularly by placing Chlamydomonas in the context of other algae. Is it better or worse? What advantages and limitations does it have?

-Chlamydomonas consortia are critical in many different aspects, including hydrogen production, therefore It is also important to mention that algae-bacteria consortia are being utilized to enhance the efficiency of hydrogen production, especially as shown recently with nitrogen-fixing bacteria.

-I miss a table that at least indicates the most recent studies, including the mutated or overexpressed genes, the genetic background of Chlamydomonas, the physiological condition of production, and the yield obtained.

-L215: “The operation of the Q cycle  contributes to both pH and formation” I'm sorry, I don't understand what it  mean by this sentence. Please, rewrite it.

L242: “shifting of HydA activity towards H2 uptake” I'm sorry, I don't understand what it mean by " H2 uptake." Could you please clarify?

L273: “As PRK catalyses the phosphorylation of ribulose 5-phosphate by the use of ATP the amino acid change, in the absence of active PRK CBB cycle will not work” I'm sorry, I don't understand what it  mean by this sentence. Please, rewrite it.

Minors:

-Please check that the use of italics is consistent

-L415: “Click or tap here to enter text.” Tipo

-L446: “[103,104]”.? Tipo

Author Response

Reviewer 2

 We would like to thank the reviewer for the positive and critical feedback on our manuscript.

The introduction seems short to me; at the very least, it should mention that Chlamydomonas has other mechanisms of H2 production that are independent of light that are independent of PSII and can occur in the dark.

Thank you for this comment. We have expanded the introduction and added a paragraph on dark H2 production.

The introduction should be improved, particularly by placing Chlamydomonas in the context of other algae. Is it better or worse? What advantages and limitations does it have?

In regard to photo-H2 production and mechanistic detailed work, no other comparable algal system has been developed.

Chlamydomonas consortia are critical in many different aspects, including hydrogen production, therefore It is also important to mention that algae-bacteria consortia are being utilized to enhance the efficiency of hydrogen production, especially as shown recently with nitrogen-fixing bacteria.

The focus of this review is on light-driven H2 production. We agree, that algae-bacteria consortia are an interesting topic, yet, we feel, that this would be out of the scope of the current review.

I miss a table that at least indicates the most recent studies, including the mutated or overexpressed genes, the genetic background of Chlamydomonas, the physiological condition of production, and the yield obtained.

A table, summarizing key proteins affecting light-driven H2 production in Chlamydomonas reinhardtii was added.

L215: “The operation of the Q cycle contributes to both pH and formation” I'm sorry, I don't understand what it  mean by this sentence. Please, rewrite it.

It was changed to “The operation of the Q cycle [72] contributes to both DpH and Dψ formation.”

L242: “shifting of HydA activity towards H2 uptake” I'm sorry, I don't understand what it mean by " H2 uptake." Could you please clarify?

HydA may also take up H2 to reduce ferredoxin.

L273: “As PRK catalyses the phosphorylation of ribulose 5-phosphate by the use of ATP the amino acid change, in the absence of active PRK CBB cycle will not work” I'm sorry, I don't understand what it  mean by this sentence. Please, rewrite it.

Thanks! This sentence was rewritten.

Minors: 

-Please check that the use of italics is consistent

-L415: “Click or tap here to enter text.” Tipo

-L446: “[103,104]”.? Tipo

Done.

Round 2

Reviewer 2 Report

Comments and Suggestions for Authors

I believe the authors have adequately addressed all of my comments and suggestions, and I accept the paper in its current version.